# PIDForest: Anomaly Detection via Partial Identification

**Parikshit Gopalan**
VMware Research
pgopalan@vmware.com

**Vatsal Sharan**
Stanford University
vsharan@stanford.edu

**Udi Wieder**
VMware Research
uwieder@vmware.com

## Abstract

We consider the problem of detecting anomalies in a large dataset. We propose a framework called Partial Identification which captures the intuition that anomalies are easy to distinguish from the overwhelming majority of points by relatively few attribute values. Formalizing this intuition, we propose a geometric anomaly measure for a point that we call PIDScore, which measures the minimum density of data points over all subcubes containing the point. We present PIDForest: a random forest based algorithm that finds anomalies based on this definition. We show that it performs favorably in comparison to several popular anomaly detection methods, across a broad range of benchmarks. PIDForest also provides a succinct explanation for why a point is labelled anomalous, by providing a set of features and ranges for them which are relatively uncommon in the dataset.[1]

## 1 Introduction

An anomaly in a dataset is a point that does not conform to what is normal or expected. Anomaly detection is a ubiquitous machine learning task with diverse applications including network monitoring, medicine and finance [1, Section 3]. There is an extensive body of research devoted to it, see [1, 2] and the references therein. Our work is primarily motivated by the emergence of large distributed systems, like the modern data center which produce massive amounts of heterogeneous data. Operators need to constantly monitor this data, and use it to identify and troubleshoot problems. The volumes of data involved are so large that a lot of the analysis has to be automated. Here we highlight some of the challenges that an anomaly detection algorithm must face.

1. **High dimensional, heterogeneous data:** The data collected could contains measurements of metrics like cpu usage, memory, bandwidth, temperature, in addition to categorical data such as day of the week, geographic location, OS type. This makes finding an accurate generative model for the data challenging. The metrics might be captured in different units, hence algorithms that are unit-agnostic are preferable. The algorithm needs to scale to high dimensional data.

2. **Scarce labels:** Most of the data are unlabeled. Generating labels is time and effort intensive and requires domain knowledge. Hence supervised methods are a non-starter, and even tuning too many hyper-parameters of unsupervised algorithms could be challenging.

3. **Irrelevant attributes:** Often an anomaly manifests itself in a relatively small number of attributes among the large number being monitored. For instance, a single machine in a large datacenter might be compromised and behave abnormally.

4. **Interpretability of results:** When we alert a datacenter administrator to a potential anomaly, it helps to point to a few metrics that might have triggered it, to help in troubleshooting.

In the generative model setting, anomalies come with a simple explanation: a model that fits the data, under which the anomalous observation is unlikely. Interpretability is more challenging for algorithms that do not assume a generative model. In this work, we are particularly interested in random forest based methods for anomaly detection, namely the influential work on Isolation Forests [3] (we refer to this algorithm as iForest) and subsequent work [4, 5]. iForest is a remarkably simple and efficient algorithm, that has been found to outperform other anomaly detection methods in several domains [6]. Yet, there is no crisp definition of ground truth for what constitutes an anomaly: the anomaly score is more or less *defined* as the output of the algorithm. We believe that a necessary step for interpretability is a clear articulation of *what* is an anomaly, separate from the algorithmic question of *how* it is found.

**Our contributions.** We summarize the main contributions of this work:

1. In Section 2, we motivate and propose a new anomaly measure that we call PIDScore. Our definition corresponds to an intuitive notion of what is an anomaly and has a natural geometric interpretation. It is inspired by the notion of Partial Identification introduced by Wigderson and Yehudayoff [7] can be viewed as a natural generalization of teaching dimension [8, 9].

2. Our definition sheds light on the types of points likely to be labeled as anomalies by the iForest algorithm, and also on the types of points it might miss. We build on this intuition to design an efficient random forest based algorithm—PIDForest, which finds anomalies according to PIDScore, in Section 3.

3. We present extensive experiments on real and synthetic data sets showing that our algorithm consistently outperforms or matches six popular anomaly detection algorithms. PIDForest is the top performing algorithm in 6 out of 12 benchmark real-world datasets, while no other algorithm is the best in more than 3. PIDForest is also resilient to noise and irrelevant attributes. These results are in Section 4 and 5.

We begin by describing our proposed anomaly measure, PIDScore at a high level. Let the *sparsity* of a dataset $\mathcal{T}$ in a subcube of the attribute space be the volume of the subcube divided by the number of points from $\mathcal{T}$ that it contains. For a dataset $\mathcal{T}$ and a point $x$, $\mathrm{PIDScore}(x, \mathcal{T})$ measures the maximum sparsity of $\mathcal{T}$ in all subcubes $C$ containing $x$. A point $x$ is labelled anomalous if it belongs to a region of the attribute space where data points are sparse.

Given this definition, one could aim for an algorithm that preprocesses $\mathcal{T}$, then takes a point $x$ and computes $\mathrm{PIDScore}(x, \mathcal{T})$. Such an algorithm is likely to suffer from the curse of dimensionality like in Nearest Neighbor based methods, and not scale to high volumes of data. Instead we adopt the approach of iForest [3] which focuses on what is anomalous, rather than the entire dataset. We call the resulting algorithm PIDForest. Like in iForest, PIDForest builds a collection of decision trees that partition space into subcubes. In PIDForest, the choice of the splits at each node favors partitions of greatly varying sparsity, the variance in the sparsity is explicitly the quantity we optimize when choosing a split. In contrast, previous work either choose splits randomly [3] or based on the range [4]. Choosing coordinates that have greater variance in their marginal distribution lets us hone in on the important coordinates, and makes our algorithm robust to irrelevant/noisy attributes, which are unlikely to be chosen. Secondly, we label each leaf by its sparsity rather than depth in the tree. The score of a point is the maximum sparsity over all leaves reached in the forest.

While notions of density have been used in previous works on clustering and anomaly detection, our approach differs from prior work in important ways.

1. **Dealing with heterogeneous attributes:** Dealing with subcubes and volumes allows us to handle heterogeneous data where some columns are real, some are categorical and possibly unordered. All we need is to specify two things for each coordinate: what constitutes an interval, and how length is measured. Subcubes and volumes are then defined as products over coordinates. This is in sharp contrast to methods that assume a metric. Notions like $\ell_1/\ell_2$ distance add different coordinates and might not be natural in heterogeneous settings.

2. **Scale invariance:** For a subcube, we only care about the ratio of its volume to the volume of the entire attribute space. Hence we are not sensitive to the units of measurement.

3. **Considering subcubes at all scales:** In previous works, density is computed using balls of a fixed radius, this radius is typically a hyperparameter. This makes the algorithm susceptible

to masking, since there may be a dense cluster of points, all of which are anomalous. We take the minimum over subcubes at all scales.

**Organization.** We present the definition of PIDScore in Section 2, and the PIDForest algorithm (with a detailed comparison to iForest) in Section 3. We present experiments on real world data in Section 4 and synthetic data in Section 5. We further discuss related work in Section A in the Appendix (which is included in the supplementary material).

## 2 Partial Identification and PIDScore

**A motivating example: Anomalous Animals.** Imagine a tabular data set that contains a row for every animal on the planet. Each row then contains attribute information about the animal such as the species, color, weight, age and so forth. The rows are ordered. Say that Alice wishes to identify a particular animal in the table unambiguously to Bob, using the fewest number of bits.

If the animal happens to be a *white elephant*, then Alice is in luck. Just specifying the species and color narrows the list of candidates down to about fifty (as per Wikipedia). At this point, specifying one more attribute like weight or age will probably pin the animal down uniquely. Or she can just specify its order in the list.

If the animal in question happens to be a *white rabbit*, then it might be far harder to uniquely identify, since there are tens of millions of white rabbits, unless that animal happens to have some other distinguishing features. Since weight and age are numeric rather than categorical attributes, if one could measure them to arbitrary precision, one might be able to uniquely identify each specimen. However, the higher the precision, the more bits Alice needs to communicate to specify the animal.

We will postulate a formal definition of anomaly score, drawing on the following intuitions:

1. **Anomalies have short descriptions.** The more exotic/anomalous the animal Alice has in mind, the more it stands out from the crowd and the easier it is for her to convey it to Bob. Constraining just a *few carefully chosen* attributes sets anomalies apart from the vast majority of the population.
2. **Precision matters in real values.** For real-valued attributes, it makes sense to specify a range in which the value lies. For anomalous points, this range might not need to be very narrow, but for normal points, we might need more precision.
3. **Isolation may be overkill.** The selected attributes need not suffice for complete isolation. *Partial identification* aka narrowing the space down to a small list can be a good indicator of an anomaly.

First some notation: let $\mathcal{T}$ denote a dataset of $n$ points in $d$ dimensions. Given indices $S \subseteq [d]$ and $x \in \mathbb{R}^d$, let $x_S$ denote the projection of $x$ onto coordinates in $S$. Logarithms are to base 2.

### 2.1 The Boolean setting

We first consider the Boolean setting where the set of points is $\mathcal{T} \subseteq \{0, 1\}^d$. Assume that $\mathcal{T}$ has no duplicates. We define idLength$(x, \mathcal{T})$ to be the minimum number of co-ordinates that must be revealed to uniquely identify $x$ among all points in $\mathcal{T}$. Since there are no duplicates, revealing all coordinates suffices, so idLength$(x, \mathcal{T}) \leq d$,

**Definition 1.** (IDs for a point) We say that $S \subseteq [d]$ is an ID for $x \in \mathcal{T}$ if $x_S \neq y_S$ for all $y \in \mathcal{T} \setminus \{x\}$. Let ID$(x, \mathcal{T})$ be the smallest ID for $x$ breaking ties arbitrarily. Let idLength$(x, \mathcal{T}) = |\text{ID}(x, \mathcal{T})|$.

While on first thought idLength is an appealing measure of anomaly, it does not deal with duplicates, and further, the requirement of unique identification is fairly stringent. Even in simple settings points might not have short IDs. For example, if $\mathcal{H}$ is the Hamming ball consisting of $0^d$ and all $d$ unit vectors, then idLength$(0^d, \mathcal{H}) = d$, since we need to reveal all the coordinates to separate $0^d$ from every unit vector. One can construct examples where even the average value of idLength$(x, \mathcal{T})$ over all points can be surprisingly high [9].

We relax the definition to allow for *partial identification*. Given $x \in \mathcal{T}$ and $S \subseteq [d]$, the set of impostors of $x$ in $\mathcal{T}$ are all points that equal $x$ on all coordinates in $S$. Formally $\text{Imp}(x, \mathcal{T}, S) = \{y \in \mathcal{T} \text{ s.t. } x_S = y_S\}$. We penalize sets that do not identify $x$ uniquely by the *logarithm of the number of impostors*. The intuition is that this penalty measures how many bits it costs Alice to specify $x$ from the list of impostors.

**Definition 2.** (Partial ID) We define

$$\mathrm{PID}(x, \mathcal{T}) = \arg \min_{S \subseteq [d]} (|S| + \log_2(|\mathrm{Imp}(x, \mathcal{T}, S)|)), \qquad (1)$$

$$\mathrm{pidLength}(x, \mathcal{T}) = \min_{S \subseteq [d]} (|S| + \log_2(|\mathrm{Imp}(x, \mathcal{T}, S)|)). \qquad (2)$$

It follows from the definition that $\mathrm{pidLength}(x, \mathcal{T}) \leq \min(\log_2(n), \mathrm{idLength}(x, \mathcal{T}))$. The first inequality follows by taking $S$ to be empty so that every point in $\mathcal{T}$ is an impostor, the second by taking $S = \mathrm{ID}(x, \mathcal{T})$ so that the only impostor is $x$ itself. Returning to the Hamming ball example, it follows that $\mathrm{pidLength}(0^d, \mathcal{T}) = \log_2(d + 1)$ where we take the empty set as the PID.

We present an alternate geometric view of $\mathrm{pidLength}$, which generalizes naturally to other settings. A subcube $C$ of $\{0, 1\}^d$ is the set of points obtained by fixing some subset $S \subseteq [d]$ coordinates to values in $0, 1$. The sparsity of $\mathcal{T}$ in a subcube $C$ is $\rho_{0,1}(\mathcal{T}, C) = |C|/|C \cap \mathcal{T}|$. The notation $C \ni x$ means that $C$ contains $x$, hence $\min_{C \ni x}$ is the minimum over all $C$ that contain $x$. One can show that for $x \in \mathcal{T}$, $\max_{C \ni x} \rho_{0,1}(\mathcal{T}, C) = 2^{d - \mathrm{pidLength}(x, \mathcal{T})}$, see appendix D for a proof. This characterization motivates using $2^{-\mathrm{pidLength}(x, \mathcal{T})}$ as an anomaly score: anomalies are points that lie in relatively sparse subcubes. Low scores come with a natural witness: a sparse subcube $\mathrm{PID}(x, \mathcal{T})$ containing relatively few points from $\mathcal{T}$.

## 2.2 The continuous setting

Now assume that all the coordinates are real-valued, and bounded. Without loss of generality, we may assume that they lie in the range $[0, 1]$, hence $\mathcal{T}$ is a collection of $n$ points from $[0, 1]^d$. An interval $I = [a, b], 0 \leq a \leq b \leq 1$ is of length $\mathrm{len}(I) = b - a$. A subcube $C$ is specified by a subset of co-ordinates $S$ and intervals $I_j$ for each $j \in S$. It consists of all points such that $x_j \in I_j$ for all $j \in S$. To simplify our notation, we let $C$ be $I_1 \times I_2 \cdots \times I_d$ where $I_j = [0, 1]$ for $j \notin S$. Note that $\mathrm{vol}(C) = \Pi_j \mathrm{len}(I_j)$. Define the sparsity of $\mathcal{T}$ in $C$ as $\rho(\mathcal{T}, C) = \mathrm{vol}(C)/|C \cap \mathcal{T}|$. $\mathrm{PIDScore}(x, T)$ is the maximum sparsity over all subcubes of $[0, 1]^d$ containing $x$.

**Definition 3.** For $x \in \mathcal{T}$, let

$$\mathrm{PID}(x, \mathcal{T}) = \arg \max_{C \ni x} \rho(\mathcal{T}, C), \quad \mathrm{PIDScore}(x, \mathcal{T}) = \max_{C \ni x} \rho(\mathcal{T}, C).$$

To see the analogy to the Boolean case, define $\mathrm{pidLength}(x, \mathcal{T}) = -\log(\mathrm{PIDScore}(x, \mathcal{T}))$. Fix $C = \mathrm{PID}(x, \mathcal{T})$. Since $\mathrm{vol}(C) = \prod_{j \in [d]} \mathrm{len}(I_j)$, we can write

$$\mathrm{pidLength}(x, \mathcal{T}) = \log(|C \cap \mathcal{T}|/\mathrm{vol}(C)) = \sum_{j \in [d]} \log(1/\mathrm{len}(I_j)) + \log(|C \cap \mathcal{T}|). \qquad (3)$$

This exposes the similarities to Equation (2): $C \cap \mathcal{T}$ is exactly the set of impostors for $x$, whereas $\sum_{j \in [d]} \log(1/\mathrm{len}(I_j))$ is the analog of $|S|$. In the boolean setting, we pay 1 for each coordinate from $S$, here the cost ranges in $[0, \infty)$ depending on the length of the interval. In the continuous setting, the $j \notin S$ iff $I_j = [0, 1]$ hence $\log(1/\mathrm{len}(I_j)) = 0$, hence we pay nothing for coordinates outside $S$. Restricting to an interval of length $p$ costs $\log(1/p)$. If $p = 1/2$, we pay 1, which is analogous to the Boolean case where we pay 1 to cut the domain in half. This addresses the issue of having to pay more for higher precision. Note also that the definition is *scale-invariant* as multiplying a coordinate by a constant changes the volume of all subcubes by the same factor.

**Other attributes:** To handle attributes over a domain $D$, we need to specify what subsets of $D$ are intervals and how we measure their length. For discrete attributes, it is natural to define $\mathrm{len}(I) = |I|/|D|$. When the domain is ordered intervals are naturally defined, for instance *months between April and September* is an interval of length $1/2$. We could also allow wraparound in intervals, say *months between November and March*. For unordered discrete values, the right definition of interval could be singleton sets, like *country = Brazil* or certain subsets, like *continent = the Americas*. The right choice will depend on the dataset. Our definition is flexible enough to handle this: We can make independent choices for each coordinate, subcubes and volumes are then defined as products, and PIDScore can be defined using definition 3.

**IDs and PIDs.** The notion of IDs for a point is natural and has been studied in the computational learning literature under various names: the teaching dimension of a hypothesis class [8], discriminant

[10], specifying set [11] and witness set [9]. Our work is inspired by the work of Wigderson and Yehudayoff [7] on population recovery, which is the task of learning mixtures of certain discrete distributions on the Boolean cube. Their work introduces the notion of partial identification in the Boolean setting. The terminology of IDs and impostors is from their work. They also consider PIDs, but with a different goal in mind (to minimize the depth of a certain graph constructed using the PID relation). Our definition of $\mathrm{pidLength}$, the extension of partial identification to the continuous setting and its application to anomaly detection are new contributions.

## 3 The PIDForest algorithm

We do not know how to compute PIDScore exactly, or even a provable approximation of it in a way that scales well with both $d$ and $n$. The PIDForest algorithm described below is heuristic designed to approximate PIDScore. Like with iForest, the PIDForest algorithm builds an ensemble of decision trees, each tree is built using a sample of the data set and partitions the space into subcubes. However, the way the trees are constructed and the criteria by which a point is declared anomalous are very different. Each node of a tree corresponds to a subcube $C$, the children of $C$ represent a disjoint partition of $C$ along some axis $i \in [d]$ (iForest always splits $C$ into two , here we allow for a finer partition). The goal is to have large variance in the sparsity of the subcubes. Recall that the sparsity of a subcube $C$ with respect to a data set $\mathcal{T}$ is $\rho(C, \mathcal{T}) = \mathrm{vol}(C)/|C \cap \mathcal{T}|$. Ultimately, the leaves with large $\rho$ values will point to regions with anomalies.

For each tree, we pick a random sample $\mathcal{P} \subseteq \mathcal{T}$ of $m$ points, and use that subset to build the tree. Each node $v$ in the tree corresponds to subcube $C(v)$, and a set of points $P(v) = C(v) \cap \mathcal{P}$. For the root, $C(v) = [0,1]^d$ and $P(v) = \mathcal{P}$. At each internal node, we pick a coordinate $j \in [d]$, and breakpoints $t_1 \leq \cdots \leq t_{k-1}$ which partition $I_j$ into $k$ intervals, and split $C$ into $k$ subcubes. The number of partitions $k$ is a hyper-parameter (taking $k < 5$ works well in practice). We then partition the points $P(v)$ into these subcubes. The partitions stop when the tree reached some specified maximum depth or when $|P(v)| \leq 1$. The key algorithmic problem is how to choose the coordinate $j$ and the breakpoints by which it should be partitioned. Intuitively we want to partition the cube into some sparse regions and some dense regions. This intuition is formalized next.

Let $I_j \subseteq [0,1]$ be the projection of $C$ onto coordinate $i$. Say the breakpoints are chosen so that we partition $I_j$ into $I_j^1, \ldots, I_j^k$. This partitions $C$ into $C^1, \ldots, C^k$ where the intervals corresponding to the other coordinates stay the same. We first observe that in any partition of $C$, the *average* sparsity of the subcubes when weighted by the number of points is the same. Let

$$p_i := \frac{\mathrm{len}(I_j^i)}{\mathrm{len}(I)} = \frac{\mathrm{vol}(C^i)}{\mathrm{vol}(C)}, q_i := \frac{|P \cap C^i|}{|P|},$$

$$\implies \rho(C^i) = \frac{\mathrm{vol}(C^i)}{|P \cap C^i|} = \frac{p_i \mathrm{vol}(C)}{q_i |P|} = \frac{p_i \rho(C)}{q_i}.$$

Since a $q_i$ fraction of points in $P$ have sparsity $\rho(C_i)$, the expected sparsity for a randomly chosen point from $P$ is

$$\sum_{i \in [k]} q_i \rho(C^i) = \sum_{i \in [k]} p_i \rho(C) = \rho(C).$$

In words, in any partition of $C$ if we pick a point randomly from $P(v)$ and measure the sparsity of its subcube, on expectation we get $\rho(C)$. Recall that our goal is to split $C$ into sparse and dense subcubes. Hence a natural objective is to maximize the *variance* in the sparsity:

$$\mathbf{Var}(\mathcal{P}, \{C^i\}_{i \in [k]}) = \sum_{i \in [k]} q_i (\rho(C^i) - \rho(C))^2 = \sum_{i \in [k]} q_i \rho(C^i)^2 - \rho(C)^2. \tag{4}$$

A partition that produces large variance in the sparsity needs to partition space into some sparse regions and other dense regions, which will correspond to outliers and normal regions respectively. Alternately, one might choose partitions to optimize the maximum sparsity of any interval in the partition, or some higher moment of the sparsity. Maximizing the variance has the advantage that it turns out to equivalent to a well-studied problem about histograms, and admits a very efficient streaming algorithm. We continue splitting until a certain predefined depth is reached, or points are isolated. Each leaf is labeled with the sparsity of its subcube.

```
PIDForest Fit
Params:  Num of trees t, Samples m, Max degree k, Max depth h.
Repeat t times:
    Create root node v.
    Let C(v) = [0,1]^d, P(v) ⊆ T be a random subset of size m .
    Split(v)

Split(v):
    For j ∈ [d], compute the best split into k intervals.
    Pick j that maximizes variance, split C along j into {C^i}_{i=1}^k.
    For i ∈ [k] create child v_i s.t.  C(v_i) = C^i, P(v_i) = P(v) ∩ C^i.
    If depth(v_i) ≤ h and |P(v_i)| > 1 then Split(v_i).
    Else, set PIDScore(v_i) = vol(C(v_i))/|P(v_i)|.
```

In Appendix C, we show that the problem of finding the partition along a coordinate $j$ that maximizes the variance in the sparsity can be reduced to the problem of finding a $k$-histogram for a discrete function $f : [n] \to \mathbb{R}$, which minimizes the squared $\ell_2$ error. This is a well-studied problem [12, 13, 14] and there an efficient one-pass streaming algorithm for computing near-optimal histograms due to Guha *et al.* [15]. We use this algorithm to compute the best split along each coordinate, and then choose the coordinate that offers the most variance reduction. Using their algorithm, finding the optimal split for a node takes time $O(dm \log(m))$. This is repeated at most $k^h$ times for each tree (typically much fewer since the trees we build tend to be unbalanced), and $t$ times to create the forest. We typically choose $m \le 200$, $k \le 5$, $h \le 10$ and $t \le 50$.

Producing an anomaly score for each point is fairly straightforward. Say we want to compute a score for $y \in [0,1]^d$. Each tree in the forest maps $y$ to a leaf node $v$ and gives it a score PIDScore($v$). We take the 75% percentile score as our final score (any robust analog of the max will do).

**Comparison to Isolation Forest.**   iForest repeatedly samples a set $S$ of $m$ points from $\mathcal{T}$ and builds a random tree with those points as leaves. The tree is built by choosing a random co-ordinate $x_i$, and a random value in its range about which to split. The intuition is that anomalous points will be easy to separate from the rest, and will be isolated at small depth. What kind of points are likely to be labeled anomalous by iForest?

In one direction, if a point is isolated at relatively low depth $k$ in a tree, then it probably belongs in a sparse subcube. Indeed, a node at depth $k$ corresponds to a subcube $C$ of expected volume $2^{-k}$, which is large for small $k$. The fact that the sample contains no points from $C$ implies that $C \cap \mathcal{T}$ is small, with high probability (this can be made precise using a VC-dimension argument). Hence $\rho(C,T) = \text{vol}(C)/|C \cap \mathcal{T}|$ is fairly large.

Being in a sparse subcube is necessary but not sufficient. This is because iForest chooses which coordinate we split on as well as the breakpoint at random. Thus to be isolated at small depth frequently, a point needs to lie in an *abundant* number of low-density subspaces: picking splits at random should have a good chance of defining such a subspace. Requiring such an abundance of sparse subcubes can be problematic. Going back to the animals example, isolating white elephants is hard unless both Color and Type are used as attributes, as there is no shortage of elephants or white animals. Moreover, which attributes are relevant can depend on the point: weight might be irrelevant in isolating a white elephant, but it might be crucial to isolating a particularly large elephant. This causes iForest to perform poorly in the presence of irrelevant attributes, see for instance [5].

This is the fundamental difference between PIDForest and iForest and its variants. PIDForest zooms in on coordinates with signal—where a split is most beneficial. Attributes with little signal are unlikely to be chosen for splitting. For concrete examples, see Section 5. The tradeoff is that we incur a slightly higher cost at training time, the cost of prediction stays pretty much the same.

## 4   Real-world Datasets

We show that PIDForest performs favorably in comparison to several popular anomaly detection algorithms on real-world benchmarks. We select datasets from varying domains, and with different

number of datapoints, percentage of anomalies and dimensionality. The code and data for all experiments is available online.[2] Detailed parameters of the datasets are in Table 2 in the appendix.

**Dataset Descriptions:** The first set of datasets are classification datasets from the UCI [16] and openML repository [17] (they are also available at [18]). Three of the datsets—*Thyroid*, *Mammography* and *Seismic*—are naturally suited to anomaly detection as they are binary classification tasks where one of the classes has a much smaller occurrence rate (around $5\%$) and hence can be treated as anomalous. *Thyroid* and *Mammography* have real-valued attributes whereas *Seismic* has categorical attributes as well. Three other datasets—*Satimage-2*, *Musk* and *Vowels*—are classification datasets with multiple classes, and we combine the classes and divide them into inliers and outliers as in [19]. Two of the datasets—*http* and *smtp*—are derived from the KDD Cup 1999 network intrusion detection task and we preprocess them as in [20]. These two datasets have have significantly more datapoints (about 500k and 100k respectively) and a smaller percentage of outliers (less than $0.5\%$).

The next set of real-world datasets—*NYC taxicab*, *CPU utilization*, *Machine temperature (M.T.)* and *Ambient temperature (A.T.)*—are time series datasets from the Numenta anomaly detection benchmark which have been hand-labeled with anomalies rooted in real-world causes [21]. The length of the time series is 10k-20k, with about $10\%$ of the points marked anomalous. We use the standard technique of *shingling* with a sliding window of width 10, hence each data point becomes a 10 dimensional vector of 10 consecutive measurements from the time series.

**Methodology:** We compare PIDForestwith six popular anomaly detection algorithms: Isolation Forest (iForest), Robust Random Cut Forest (RRCF), one-class SVM (SVM), Local Outlier Factor (LOF), k-Nearest Neighbour (kNN) and Principal Component Analysis (PCA). We implement PIDForest in Python, it takes about 500 lines of code. For iForest, SVM and LOF we used the scikit-learn implementations, for kNN and PCA we used the implementations on PyOD [22] , and for RRCF we use the implementation from [23]. Except for RRCF, we run each algorithm with the default hyperparameter setting as varying the hyperparameters from their default values did not change the results significantly. For RRCF, we use 500 trees instead of the default 100 since it yielded significantly better performance. For PIDForest, we fix the hyperparameters of depth to 10, number of trees to 50, and the number of samples used to build each tree to 100. We use the area under the ROC curve (AUC) as the performance metric. As iForest, PIDForest and RRCF are randomized, we repeat these algorithms for 5 runs and report the mean and standard deviation. SVM, LOF, kNN and PCA are deterministic, hence we report a single AUC number for them.

**Results:** We report the results in Table 1. PIDForest is the top performing or jointly top performing algorithm in 6 out of the 12 datasets, and iForest, kNN and PCA are top performing or jointly top performing algorithms in 3 datasets each. Detailed ROC performance curves of the algorithms are given in Fig. 6 and 7 in the Appendix. While the running time of our fit procedure is slower than iForest, it is comparable to RRCF and faster than many other methods. Even our vanilla Python implementation on a laptop computer only takes about 5 minutes to fit a model to our largest dataset which has half a million points.

Recall from Section 3 that PIDForest differs from iForest in two ways, it optimizes for the axis to split on, and secondly, it uses sparsity instead of depth as the anomaly measure. To further examine the factors which contribute to the favorable performance of PIDForest, we do an ablation study through two additional experiments.

**Choice of split:** Optimizing for the choice of split rather than choosing one at random seems valuable in the presence of irrelevant dimensions. To measure this effect, we added 50 additional random dimensions sampled uniformly in the range $[0, 1]$ to two low-dimensional datasets from Table 1—*Mammography* and *Thyroid* (both datasets are 6 dimensional). In the *Mammography* dataset, PIDForest (and many other algorithms as well) suffers only a small $2\%$ drop in performance, whereas the performance of iForest drops by $15\%$. In the *Thyroid* dataset, the performance of all algorithms drops appreciably. However, PIDForest has a $13\%$ drop in performance, compared to a $20\%$ drop for iForest. The detailed results are given in Table 3 in the Appendix.

**Using sparsity instead of depth:** In this experiment, we test the hypothesis that the sparsity of the leaf is a better anomaly score than depth for the PIDForest algorithm. The performance of PIDForest deteriorates noticeably with depth as the score, the AUC for *Thyroid* drops to $0.847$ from $0.876$, while the AUC for *Mammography* drops to $0.783$ from $0.840$.

| Data set | PIDForest | iForest | RRCF | LOF | SVM | kNN | PCA |
|---|---|---|---|---|---|---|---|
| Thyroid | **0.876 ± 0.013** | 0.819 ± 0.013 | 0.739± 0.004 | 0.737 | 0.547 | 0.751 | 0.673 |
| Mammo. | 0.840 ± 0.010 | 0.862 ± 0.008 | 0.830 ± 0.002 | 0.720 | 0.872 | 0.839 | **0.886** |
| Seismic | **0.733 ± 0.006** | 0.698 ± 0.004 | 0.701 ± 0.004 | 0.553 | 0.601 | **0.740** | 0.682 |
| Satimage | 0.987 ± 0.001 | **0.994 ± 0.001** | **0.991 ± 0.002** | 0.540 | 0.421 | 0.936 | 0.977 |
| Vowels | 0.741 ± 0.008 | 0.736 ± 0.026 | 0.813± 0.007 | 0.943 | 0.778 | **0.975** | 0.606 |
| Musk | **1.000 ± 0.000** | **0.998 ± 0.003** | 0.998 ± 0.000 | 0.416 | 0.573 | 0.373 | **1.000** |
| http | 0.986 ± 0.004 | **1.000 ± 0.000** | 0.993 ± 0.000 | 0.353 | 0.994 | 0.231 | 0.996 |
| smtp | **0.923 ± 0.003** | 0.908 ± 0.003 | 0.886 ± 0.017 | 0.905 | 0.841 | 0.895 | 0.823 |
| NYC | 0.564 ± 0.004 | 0.550 ± 0.005 | 0.543 ± 0.004 | 0.671 | 0.500 | **0.697** | 0.511 |
| A.T. | **0.810 ± 0.005** | 0.780 ± 0.006 | 0.695 ±0.004 | 0.563 | 0.670 | 0.634 | 0.792 |
| CPU | **0.935 ± 0.003** | 0.917 ± 0.002 | 0.785 ± 0.002 | 0.560 | 0.794 | 0.724 | 0.858 |
| M.T. | 0.813 ± 0.006 | 0.828 ± 0.002 | 0.7524 ± 0.003 | 0.501 | 0.796 | 0.759 | **0.834** |

Table 1: Results on real-world datasets. We bold the algorithm(s) which get the best AUC.

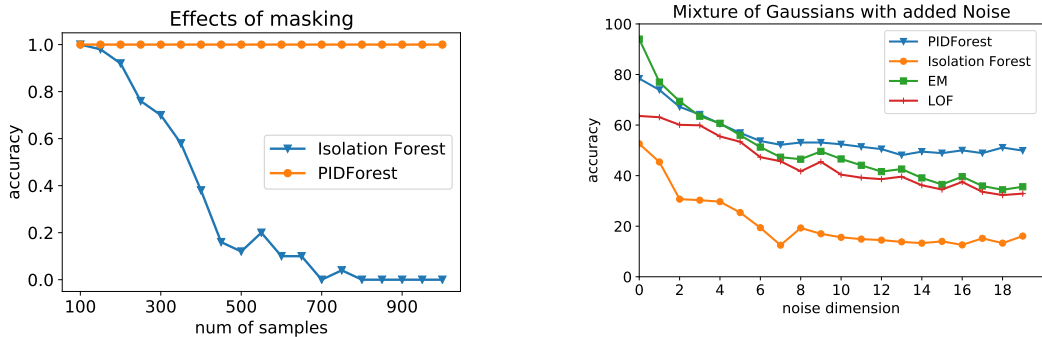

(a) Masking. Accuracy is the fraction of true outliers in the top 5% of reported anomalies.

(b) $y-$axis measures how many of the 100 true anomalies were reported by the algorithm in the top 100 anomalies.

Figure 1: Synthetic experiments on masked anomalies and Gaussian data.

## 5 Synthetic Data

We compare PIDForest with popular anomaly detection algorithms on synthetic benchmarks. The first experiment checks how the algorithms handle duplicates in the data. The second experiment uses data from a mixture of Gaussians, and highlights the importance of the choice of coordinates to split in PIDForest. The third experiment tests the ability of the algorithm to detect anomalies in time-series (see Appendix B). In all these experiments, PIDForest outperforms prior art.

**Masking and sample size.** It is often the case that anomalies repeat multiple times in the data. This phenomena is called *masking* and is a challenge for many algorithms. iForest counts on sampling to counter masking: not too many repetitions occur in the sample. But the performance is sensitive to the sampling rate, see [4, 5]. To demonstrate it, we create a data set of 1000 points in 10 dimensions. 970 of these points are sampled randomly in $\{-1, 1\}^{10}$ (hence most of these points are unique). The remaining 30 are the all-zeros vector, these constitute a masked anomaly. We test if the zero points are declared as anomalies by PIDForest and iForest under varying sample sizes. The results are reported in Fig. 1a. Whereas PIDForest consistently reported these points as anomalies, the performance of iForest heavily depends on the sample size. When it is small, then masking is negated and the anomalies are caught, however the points become hard to isolate when the sample size increases.

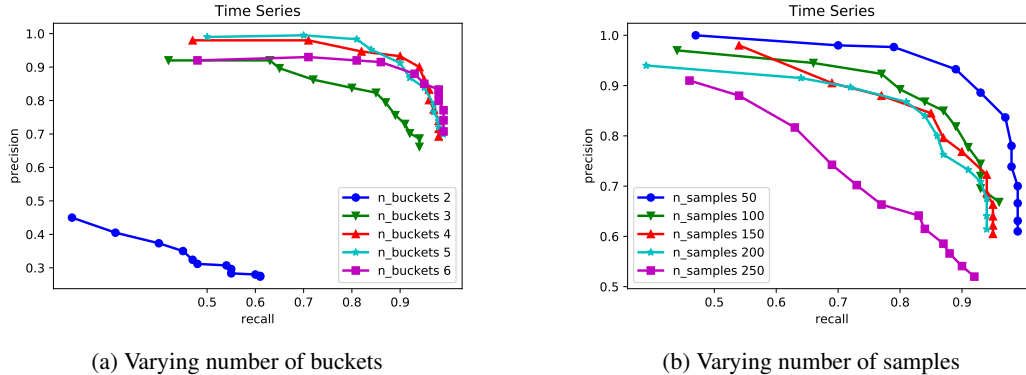

(a) Varying number of buckets          (b) Varying number of samples

Figure 2: Robustness to choice of hyperparameters

**Mixtures of Gaussians and random noise.** We use a generative model where the ground truth anomalies are the points of least likelihood. The first two coordinates of the 1000 data points are sampled by taking a mixture of two $2-$dimensional Gaussians with different means and covariances (the eigenvalues are $\{1, 2\}$, the eigenvectors are chosen at random). The remaining $d$ dimensions are sampled uniformly in the range $[-2, 2]$. We run experiments varying $d$ from 0 to 19. In each case we take the 100 points smallest likelihood points as the ground truth anomalies. For each algorithm we examine the 100 most anomalous points and calculate how many of these belong to the ground truth. We compare PIDForest with Isolation Forest, Local Outlier Factor (LOF), an algorithm that uses Expectation Maximization (EM) to fit the data to a mixture of Gaussians, RRCF, SVM, kNN and PCA. For clarity, the results for the first four algorithms are reported in Fig. 1b and the rest appear in Fig. 4b in the Appendix. Note that even without noise ($d = 0$) PIDForest is among the best generic algorithms. As the number of noisy dimensions increase PIDForest focuses on the dimensions with signal, so it performs better. Some observations:

1. The performance of iForest degrades rapidly with $d$, once $d \geq 6$ it effectively outputs a random set. Noticeably, PIDForest performs better than iForest even when $d = 0$. This is mainly due to the points between the two centers being classified as normal by iForest. PIDForest classifies them correctly as anomalous even though they are assigned to leaves that are deep in the tree.
2. The EM algorithm is specifically designed to fit a mixture of two Gaussians, so it does best for small or zero noise, i.e. when $d$ is small. PIDForest beats it once $d > 2$.
3. Apart from PIDForest, a few other algorithms such as RRCF, SVM and kNN also do well (see Fig. 4b in the Appendix)—but their performance is crucially dependent on the fact that the noise is of the same scale as the Gaussians. If we change the scale of the noise (which could happen if the measuring unit changes), then the performance of all algorithms except PIDForest drops significantly. In Figs. 5a and 5b in the Appendix, we repeat the same experiment as above but with the noisy dimensions being uniform in $[-10, 10]$ (instead of $[-2, 2]$). The performance of PIDForest is essentially unchanged, and it does markedly better than all other algorithms.

**Robustness to choice of hyperparameters.** One of the appealing properties of PIDForest is that the quality of the output is relatively insensitive to the exact value of the hyper-parameters. We tested multiple settings and found that each hyper-parameter has a moderate value above which the quality of the output remains stable. Figure 2a shows precision-recall in the synthetic time-series experiment, where we vary the parameter $k$ (number of buckets). We see that $k = 2$ is too little, but there is relatively small variation between $k = 3, 4, 5, 6$. Similar behavior was observed for the number of samples $m$ (see Figure 2b), number of trees $t$ and depth of each tree $h$, and also with the mixture of Gaussians experiment. Since these parameters do affect the running time directly, we set them to the smallest values for which we got good results.

## 6 Conclusion

We believe that PIDForest is arguably one of the best off-the-shelf algorithms for anomaly detection on a large, heterogenous dataset. It inherits many of the desirable features of Isolation Forests, while also improving on it in important ways. Developing provable and scalable approximations to PIDScore is an interesting algorithmic challenge.

## Acknowledgment

VS's contribution were supported by NSF award 1813049.

## Footnotes

[1]The full version of this paper is available at https://arxiv.org/abs/1912.03582.

[2] https://github.com/vatsalsharan/pidforest

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
