[Supplementary Material]

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

## A  Relation to prior work

Anomaly detection is a widely studied area with a number of comprehensive surveys and books [1, 24, 25]. In the following we discuss the basic techniques and how they relate to our work.

**Proximity based methods:** Many works detect anomalies by computing distances to nearest neighbors in various methods [5, 26, 27, 28, 29]. There are many different methods with pros and cons, which we do not discuss here. The important common feature is that these algorithms utilize a notion of *distance*, typically Euclidean, as a proxy for similarly across points. Once the notion of distance is established it is used find the closest neighbor, or the close $k$ neighbors, it is used to define a ball of a given radius around a point and so on.

There are few major drawbacks with these approaches. First, as the number of dimensions increases the distances between points become more and more similar, and notion of a local neighborhood looses its meaning. Further, distance is an ill suited notion to the general case where some columns are categorical and some numeric and where we different columns employ different units. Isolation based algorithms work with the input basis, they do not compute linear combinations of attributes which are required to change basis. This is an advantage in dealing with heterogeneous data.

Having said that, for homogeneous datasets that involve audio or visual inputs (like Vowels/MNIST), the input basis may not be the *right* basis. If the signal has sparsity in some special basis, then $\ell_2$ distance, or some variant of it like Mahalanobis distance might be a natural metric for such settings. In such situations kNN, PCA might do better than isolation based approaches.

**Density based algorithms:** Density has been used as a criterion for several works on clustering. For instance, DBSCAN [30] builds clusters from maximal connected components out of dense balls (in a chosen metric) of a certain radius. Outliers are points that do not belong in any such cluster. The work of Agrawal *et al.* [31] builds clusters that can be expressed as connected unions of subcubes. A key difference from these works is that we do not attempt to discover the structure (such as clusters) in the normal data. Further, rather than only consider balls/subcubes at a particular scale (this scale is a hyperparameter), our algorithms attempt to minimize density over subcubes at all scales.

**Isolation Forest:** The most relevant algorithm is the widely used Isolation Forest [3], a detailed comparison of the technical differences between the two algorithms is in Section 3. Since [3] chooses *randomly* which column to split, this causes Isolation Forest's accuracy to degrade when extra dimensions are introduced (this is a well known issue, see [4, 5]). Robust Isolation Forest [4] deals with this by choosing a column based on the size of its range. This makes the algorithm scale-sensitive and results change based on the units with which the data is reported. PIDForest on the other hand zooms on the columns which have the most *signal*, which makes it robust to noisy and irrelevant dimensions. Another difference is that Isolation Forest insists on full isolation and then computes the average leaf depth, which makes it sensitive to duplicates or near duplicates. See Section 5 where we demonstrate these points via experiments.

## B  Experiments on Synthetic Time Series Data

**Time Series:** We create a periodic time series using a simple sin function with a period of 40. We choose 10 locations and fix the value for the next 20 points following each of these locations. These regions are the anomalies. Finally we add small Gaussian noise to the series. See Fig. 3a for an example. As in Section 4, we shingle the time series with a window of 10. Fig. 3b shows the ROC curve (true positive rate (TPR) vs. false positive rate (FPR)), averaged over 10 runs. Since all dimensions are a priori identical, choosing splits at random seems natural. So we expect iForest to perform well, and indeed it achieves a precision of almost 1 while catching 5 out of the 10 anomalies. However, iForest struggles to catch all anomalies, and PIDForest has a significantly better precision for high recall.

## C  Finding optimal splits efficiently

In this section we present the algorithm used to split a single dimension. The problem is easy in the discrete setting, so we focus on the continuous case. We first restate the problem we which solve.

(a) An example of a time series, the red dots represent the beginning of an anomalous segment.

(b) Performance comparison on time series data.

Figure 3: Synthetic experiments on time series data.

(a) Comparison of PIDForest with Isolation Forest, EM and LOF (same as Fig. 1b, duplicated for ease of comparison).

(b) Comparison of PIDForest with RRCF, SVM, kNN and PCA.

Figure 4: Synthetic experiments on Gaussian data (noisy dimensions are uniform in $[-2, 2]$). For clarity, we split the results into two figures. $y-$axis measures how many of the 100 true anomalies were reported by the algorithm in the top 100 anomalies.

(a) Comparison of PIDForest with Isolation Forest, EM and LOF.

(b) Comparison of PIDForest with RRCF, SVM, kNN and PCA.

Figure 5: Synthetic experiments on Gaussian data (noisy dimensions are uniform in $[-10, 10]$). For clarity, we split the results into two figures. $y-$axis measures how many of the 100 true anomalies were reported by the algorithm in the top 100 anomalies.

In what follows, we may assume the dataset $\mathcal{P} \subseteq I$ is one-dimensional, since we work with the projection of the dataset onto the dimension $j$.

For interval $I$, a $k$-interval partition of $I$ is a partition into a set $\{I_1, \ldots, I_k\}$ of disjoint intervals. Letting $q_i := |P \cap I^i|/|P|$, our goal is to maximize

$$\mathbf{Var}(\mathcal{P}, \{I^i\}_{i \in [k]}) = \sum_{i \in [k]} q_i \rho(I^i)^2 - \rho(I)^2. \tag{5}$$

Since the term $\rho(I)^2$ is independent of the partition, we can drop it and restate the problem as follows.

**Optimal $k$-split:** Given a set $\mathcal{P}$ of $m$ points $x_1 \leq \cdots \leq x_m$ from an interval $I$, and a parameter $k$, find a $k$-interval partition of $I$ so as to maximize

$$\text{cost}(\mathcal{P}, k) = \sum_{i=1}^{k} q_i \rho(I_i)^2. \tag{6}$$

By shifting and scaling, we will assume that the bounding interval $I = [0, 1]$. We also assume the $x_i$s are distinct (this is not needed, but eases notation). To simplify matters, we restrict to those intervals whose start and end points are either $e_0 = 0$, $e_m = 1$, or $e_i = (x_i + x_{i+1})/2$ for $i \in [m-1]$. This avoids issues that might arise from the precision of the end points, and from having points lie on the boundary of two intervals (to which interval should they belong?). It reduces the search space of intervals to $O(m^2)$. One can use dynamic programming to give an $O(m^2 k)$ time and $O(mk)$ space algorithm to solve the problem exactly. However this is too costly, since the procedure runs in an inner loop of PIDForest Fit. Rather we show the problem reduces to that of computing optimal $k$-histograms for an array, for which there are efficient streaming approximation algorithms known.

First some notation. An interval in $[m]$ is $J = \{i : \ell \leq i \leq u\}$, and $|J| = u - \ell + 1$. Given a function $f : [m] \to \mathbb{R}$ and an interval $J \subseteq [m]$, let $\bar{f}(J) = \sum_{i \in J} f(i)/|J|$ denote the average of $f$ over the interval $J$. A $k$-interval partition of $[m]$ is a set of pairwise disjoint intervals $\{J_1, \ldots, J_k\}$ whose union is $[m]$. Given $j \in [m]$, let $J(j) \in \mathbb{I}$ denote the interval containing it.

**Optimal $k$-histograms:** Given $f : [m] \to \mathbb{R}$, find a $k$-interval partition of $[m]$ which maximizes

$$\text{cost}(f, k) = \sum_{i \in [k]} |J_i|(\bar{f}(J_i))^2. \tag{7}$$

We show in Lemma 5 that this is equivalent to finding the $k$-histogram that minimizes the $\ell_2$-error. We now give the reduction from computing $k$-splits to $k$-histograms. Recall that $e_0 = 0$, $e_m = 1$, or $e_i = (x_i + x_{i+1})/2$ for $i \in [m-1]$. For each $i \in [m]$, let $f(i) = e_i - e_{i-1}$ denote the length of the interval $[e_{i-1}, e_i]$ which contains $x_i$. There is now a natural correspondence between the discrete interval $J_{\ell,u} = \{\ell, \cdots, u\}$ and the continuous interval $I_{\ell,u} = [e_{\ell-1}, e_u]$ which contains the points $\{x_\ell, \cdots, x_u\}$ from $\mathcal{P}$, where

$$\bar{f}(J_{\ell,u}) = \frac{\sum_{i=\ell}^{u}(e_i - e_{i-1})}{u - \ell + 1} = \frac{e_u - e_{\ell-1}}{u - \ell + 1} = \rho(I_{\ell,u})$$

Thus a $k$ interval partition of $[m]$ translates to a $k$-interval partition of $I$, with objective function

$$\text{cost}(f, k) = \sum_{i \in [k]} |J_i|(\bar{f}(J_i))^2 = \sum_{i \in [k]} |\mathcal{P} \cap I_i| \rho(I_i)^2 = m \cdot \text{cost}(\mathcal{P}, k).$$

An efficient streaming algorithm for computing approximately optimal $k$-histograms is given by Guha *et al.* [15], which requires space $O(m + k^2)$ and time $O(m + k^3)$ (we set their parameter $\varepsilon$ to 0.1). We use their algorithm in the Fit procedure to find good splits.

**Comparison with exact dynamic programming algorithm.** As discussed, the variance maximization problem also admits a much less efficient but exact dynamic programming algorithm. On a 300 dimensional array, the algorithm of [15] is 50x faster than the exact dynamic program in finding a best 5-bucket histogram (0.05 s vs 2.5s). This is as expected given that the complexity of the DP is $O(m^2)$, whereas the approximate algorithm takes time $O(m)$.

# D   Proofs

**Lemma 4.** For $x \in \mathcal{T}$,

$$\max_{C \ni x} \rho_{0,1}(\mathcal{T}, C) = 2^{d - \mathrm{pidLength}(x, \mathcal{T})}.$$

**Proof:** Given $S \subseteq [d]$, let

$$C_x(S) = \{y \in \{0,1\}^d \text{ s.t. } y_S = x_S\}$$

be the subcube consisting of $2^{d-|S|}$ points that agree with $x$ on $S$. Since $C_x(S) \cap \mathcal{T} = \mathrm{Imp}(x, \mathcal{T}, S)$,

$$\rho_{0,1}(\mathcal{T}, C_x(S)) = \frac{|C_x(S)|/|C_x(S) \cap \mathcal{T}|}{|\mathcal{T}|} = \frac{2^{d-|S|}}{|\mathrm{Imp}(x, \mathcal{T}, S)|} = 2^{d - (|S| + \log_2(|\mathrm{Imp}(x, \mathcal{T}, S)|))}$$

Iterating over all subsets $S$ gives all the subcubes that contain $x$. The RHS is minimized by taking $S = \mathrm{PID}(x, \mathcal{T})$ by Definition 2. This gives the desired result. $\qquad\square$

**Lemma 5.** The $k$-histogram which maximizes the cost minimizes the squared $\ell_2$ error.

**Proof:** Given a $k$-interval partition $\{J_1, \dots, J_k\}$, the minimum $\ell_2$ error is obtained by approximating $f$ by its average over each interval. The squared $\ell_2$ errors is given by

$$
\begin{aligned}
\sum_{j \in m} (\bar{f}(J(j)) - f(j))^2 &= \sum_{j \in [m]} (f(j)^2 - 2\bar{f}(J(j))f(j) + \bar{f}(J(j))^2) \\
&= \sum_{j \in [m]} f(j)^2 - \sum_{i \in [k]} 2\bar{f}(J_i) \sum_{j \in J_i} f(j) + \sum_{i \in [k]} \sum_{j \in J_i} \bar{f}(J(j))^2) \\
&= \sum_{j \in [m]} f(j)^2 - 2 \sum_{i \in [k]} |J_i|(\bar{f}(J_i))^2 + \sum_{i \in [k]} |J_i|(\bar{f}(J_i))^2 \quad \text{since } \sum_{j \in J_i} f(j) = |J_i|\bar{f}(J_i) \\
&= \sum_{j \in J} f(j)^2 - \sum_{i \in [k]} |J_i|(\bar{f}(J_i))^2 \\
&= \sum_{j \in J} f(j)^2 - \mathrm{cost}(f, k)
\end{aligned}
$$

Hence maximizing $\mathrm{cost}(f, k)$ is equivalent to minimizing the squared error of the histogram. $\quad\square$

(a) ROC curve for *Thyroid* dataset.

(b) ROC curve for *Mammography* dataset.

(c) ROC curve for *Siesmic* dataset.

(d) ROC curve for *Satimage-2* dataset.

(e) ROC curve for *Vowels* dataset.

(f) ROC curve for *Musk* dataset.

Figure 6: ROC curves for the first six datasets from Table 1. For visual clarity, we omit LOF and SVM which did not perform as well as the other algorithms.

(a) ROC curve for *http* dataset.

(b) ROC curve for *smtp* dataset.

(c) ROC curve for *NYC taxi* dataset.

(d) ROC curve for *Ambient temperature* dataset.

(e) ROC curve for *CPU utilization* dataset.

(f) ROC curve for *Machine temperature* dataset.

Figure 7: ROC curves for the last six datasets from Table 1. For visual clarity, we omit LOF and SVM which did not perform as well as the other algorithms.

| Data set | $n$ | $d$ | #outliers (%) |
|---|---|---|---|
| Thyroid | 7200 | 6 | 534 (7.42%) |
| Mammography (Mammo.) | 11183 | 6 | 250 (2.32%) |
| Seismic | 2584 | 15 | 170 (6.5%) |
| Satimage-2 | 5803 | 36 | 71 (1.2%) |
| Vowels | 1456 | 12 | 50 (3.4%) |
| Musk | 3062 | 166 | 97 (3.2%) |
| http | 567479 | 3 | 2211 (0.4%) |
| smtp | 95156 | 3 | 30 (0.03%) |
| NYC taxicab | 10321 | 10 | 1035 (10%) |
| Ambient Temperature (A.T.) | 7267 | 10 | 726 (10%) |
| CPU utilization | 18050 | 10 | 1499 (8.3%) |
| Machine temperature (M.T.) | 22695 | 10 | 2268 (10%) |

Table 2: Details of real-world datasets. The first 8 datasets are derived from classification tasks, and the last 4 are from time series with known anomalies.

| Data set | PIDForest | iForest | RRCF | LOF | SVM | kNN | PCA |
|---|---|---|---|---|---|---|---|
| Thyroid* | **0.751 ± 0.035** | 0.641 ± 0.023 | 0.530 ± 0.005 | 0.492 | 0.494 | 0.495 | 0.614 |
| Mammography* | 0.829 ± 0.016 | 0.722 ± 0.016 | 0.797 ± 0.013 | 0.628 | **0.872** | 0.817 | 0.768 |

Table 3: For the first two datasets in Table 1 we add 50 noisy dimensions to examine the performance of algorithms in the presence of irrelevant attributes. We bold the algorithm(s) which get the best AUC, up to statistical error.