[Reviews · NeurIPS 2019]

Reviewer 1



This paper proposes a new scoring criterion (PIDScore) for the "anomalousness" of data points, based on (loosely) information-theoretic intuitions around how difficult it is to identify or characterize the point, which can be expressed in terms of the sparsity of the containing subcube. This is a richer notion that goes significantly beyond previous notions of "low density" regions for outlier detection, and also has benefits around interpretability. A corresponding algorithm (PIDForest) is developed, similar to Isolation Forests but with some differences that are motivated by PIDScore designed to overcome known weaknesses. Overall, I found the paper to be well-written and well-contextualized with respect to other work. The experiments show good performance against sound baselines, and some targeted "lesion-style" experiments support the claims around the underlying reasons why the approach works. The ideas behind the scoring function could be a good foundation for other research in anomaly detection. MORE DETAILED COMMENTS AND QUESTIONS I found the presentation of the core ideas of PIDScore (Section 2) to be clear. The differences between this approach and iForest are described and motivated well. L107: this idea gets better context in the Appendix, it might help to at least put the appropriate cites here. L42: the comparison to the "implicit" scoring definition of anomalies in iForest is very helpful for context. L74: at this point in the paper it is not very clear what this means. A simple example might help. L198: should be "in any partition _of_ C" ? L229: the parenthetical VC-dimension comment should be backed up with either a cite or clarification in the Appendix. L230: this is very well put, this paragraph captures the key distinction with iForest. L249: spelling error, "Seismic" L283: nice to see some concrete wall-clock numbers here. L289 and L297: these "lesion-style" analyses do a good job supporting the proposed claims about the benefits of the specific mechanisms of PIDForest. L313: "iForestunder" to "iForest under" L307: the synthetic experiments successfully highlight issues with other approaches. L453 (Appendix): in eq on LHS, it should be \bar{f(J(j)} (not I(J))? The Related Work section in the Appendix was informative. The Wigderson and Yehudayoff reference seems important enough to include in the main paper. The associated supplemental code submission was exceptionally thorough, it was especially helpful to browse the synthetic experiment notebooks. UPDATE: I have read the author response. The clarifications and additional detail are helpful, and if possible it would be good to slightly modify the discussions in the paper based on these. The addition of the other approaches to the "noise dimension" plot and associated discussion were also valuable, and could be included as well (perhaps in the Appendix).

Reviewer 2



This paper presents a novel outlier measure and an outlier detection algorithm to estimate the proposed measure. The tree-based measure designed can handle data sets with attributes with different nature (categorical, numerical, binary, etc.) and different scales (for numerical attributes). The authors propose a random forest-like algorithm to estimate the outlier measure. They also conduct experiments on 12 data sets where the proposed algorithm achieves the best performance on 6 data sets. The problem studied in this paper is interesting. Outlier detection is a classical task with various applications. Finding outliers in large-scale, heterogeneous data sets remains a challenging problem. One of the major contributions is the proposed outlier measure. The authors provide an outlier measure PIDScore defined based on inverse density (i.e. sparsity). They also point out how this measure unifies binary attributes and numerical attributes and achieves scale invariance for numerical attributes. I find the outlier measure definition simple yet quite intuitive. The algorithm provided is a heuristic estimate of the PIDScore. The algorithm aims to generate subcubes with large variance in terms of sparsity. However, the authors do not provide detailed explanation on why this is desired. The authors do not provide theoretical analysis on how well the algorithm can approximate the PIDScore either. Generally, the clarity of the algorithm needs improvement. The experimental results show that the algorithm outperforms baselines in some data sets but shows lower (sometimes substantially worse, e.g. on Vowels) performance on other data sets. Do authors have detailed analysis on why the algorithm does not work well on some data sets? The authors provide relatively thorough comparison against iForest, which is insightful. However, the authors may also want to include in-depth comparative analysis of the algorithm against other baselines such as PCA and KNN, especially on data sets where they achieve better performance. Detailed comments: 1) Line 17, 66, ...: heterogenous -> heterogeneous? 2) Line 168: it in -> it is 3) Line 177: We do not how -> We do not know how 4) Does the algorithm work for numerical attributes with unbounded values? %=== After Rebuttal ===% Thanks the authors for their responses. There are something in the responses that I think is worth being included in the paper. 1) I think the authors could consider to include their comparison between kNN/PCA to their methods into the final paper as it provides a more complete picture to the readers. 2) I also think it is reasonable to include some other possible options other than the proposed algorithm, and why the proposed algorithm is chosen. This can provide better understanding of the connection between the algorithm and the proposed method, and could also be inspiring for future research. 3) I suggest the authors to also investigate some earlier literature of grid-based clustering, which might be relevant. With these points addressed, I believe the paper will be more solid.

Reviewer 3



Besides the above-mentioned strong points of contributions, the following concerns should be taken into account to improve the current paper. (1) Presentation: - The Appendix is too long, but it is an optional for reviewers and readers to further read. The important contents, such as the Table 2 describing the parameters of data sets, figures of experiment results, and the related work, should not be put at the Appendix. The authors are suggested to try to make the introduction to the key idea of their approach more compact, so that these key information can be included into the main body of the paper. - The title with the keyword of "Certification" is misleading to the reviewers and readers. Actually, in the paper, there is nothing addressed that is related to how the PIDForest is doing the certification of anomaly. The authors are suggested to reconsider a more appropriate title. (2) Experiments: - In the algorithm of PIDForest, there are several parameters that will have significant impact on the performance of efficiency and accuracy. Such as $k$, the partition number would influence the size of the forest structure, which will finally influence the speed of finding optimal split for a node. These parameters $t$, $m$, $k$, and $h$ should be varying on the synthetic data sets to show their impacts on the PIDForest algorithm. - In Fig. 1 (b), the authors only showed the results of comparisons with three related methods, but indeed there are six comparative methods that are used in this paper. Why do not show the results with others, including RRCF, SVM, kNN and PCA? Is it probably that the others would have higher ability of handling random noise? Or the proposed algorithm would still show higher ability with random noise? It is not clear to reviewers and readers. This important point should be clarified in addition. - A minor concern is that the word "EM" is the first time appearing in the paper but without any descriptions. Although its meaning should be straightforward, it is still worthy making it clear explicitly for readers.

[Author Response · NeurIPS 2019]



(a) Varying number of buckets    (b) Varying number of samples    (c) Noise uniform in [-2,2]    (d) Noise uniform in [-10,10]

We thank the referees for their careful reading of the paper, their encouraging comments and thoughtful critiques. We will address all typographic and minor suggestions in the revised version. We will move the key related work and references, and more experimental results to the main body. R3 points out that we focus more on detecting anomalies rather than certifying them; we agree, and can drop the word "Certification" from the title.

**(R1) Dynamic Program vs. approximate algorithm for histograms.** On a 300 dimensional array, the approximate algorithm is 50x faster than the exact dynamic program in finding a best 5-bucket histogram (0.05 s vs 2.5s), with the approximation factor chosen to find a histogram having mean squared error at most 1.1 times the optimal histogram. This speedup is expected given that the complexity of the DP scales quadratically with array length, whereas the approximate algorithm is roughly linear. As histogram computation is in a deep inner loop and is invoked multiple times, we expect similar slowdowns when running the fit routine using the DP.

**(R2) Why is maximizing variance a good choice?** We will try to have more intuitive description of the main algorithm in the revised version. For intuition, consider the 1-dimensional case where points are generated according to a normal distribution. In this case, the outliers would be points in the tails. Indeed, if we partition the points into 3 intervals to maximize the variance in the sparsity, then we get a dense interval around the mean and two sparse intervals, one for each tail, which is exactly what we want. Moving to the general high dimensional setting, in any partition, if we pick a random point, then the expected sparsity is the same (see L202). A partition that produces large variance in the sparsity needs to partition space into some sparse regions and other dense regions, which will correspond to outliers and normal regions respectively. Alternately, one might choose partitions to optimize the maximum sparsity of any interval in the partition, or some higher moment of the sparsity. Since maximizing variance turns out to equivalent to a well-studied problem about histograms, it admits a very efficient streaming algorithm.

**(R2) Provable guarantees for PIDForest.** PIDForest is a heuristic, we do not have rigorous guarantees for it in the high-dimensional setting. This is addressed at the very start of Section 3 (L177), which states that we do not know polynomial algorithms in $n$ and $d$ that exactly compute or provably approximate PIDScore. Indeed, we suspect that exact computation might be hard as the dimension $d$ grows (and we know an exact algorithm in 1-d).

**(R2) Comparison with kNN and PCA.** Isolation based algorithms work with the input basis, they do not compute linear combinations of attributes which are required to change basis. This is an advantage in dealing with heterogenous data. But for datasets that involve audio or visual inputs (like Vowels/MNIST), the input basis may not be the *right* basis. If the signal has sparsity in some special basis, then $\ell_2$ distance (or some variant of it like Mahalanobis distance) might be a natural metric for such settings. In such situations kNN, PCA might do better. In general, anomaly detection problems are diverse and no single algorithm can be reasonably expected to be the best in every setting. The strength of PIDForest is that it makes minimal assumptions and hence works well in several settings that are common in practice and existing algorithms find challenging (e.g. heterogenous, noisy data).

**(R3) Hyperparameter settings.** One of the appealing properties of PIDForest is that the quality of the output is relatively insensitive to the exact value of the hyper-parameters. We tested multiple settings and generally found that each hyper-parameter has a moderate value above which the quality of the output doesn't change much. Figure 1a shows precision-recall in the synthetic time-series experiment, where we vary the parameter $k$ (number of buckets). We see that $k = 2$ is too little, but there isn't much difference between $k = 3, 4, 5, 6$. Similar behavior was observed for the number of samples $m$ (see Figure 1b where there is no clear trend), number of trees $t$ and depth of each tree $h$, and again with the mixture of Gaussians experiment. Since these parameters do affect the running time directly, we set them to the smallest values for which we got good results. This is how we arrive at the guidelines in L220.

**(R3) Noise Tolerance.** We have included plots for the mixtures of Gaussians experiment with additional noisy dimensions for the other algorithms (kNN, SVM, RRCF, PCA). The conclusion is that PIDForest is more noise resilient than the other algorithms. In Figure 1c we choose the noise to be uniform in $[-2, 2]$ (as in Fig 1b in the paper) and see that RRCF and PIDForest are the best algorithms. In Figure 1d, we choose the noise coordinates uniformly in $[-10, 10]$ and find that PIDForest is markedly better than RRCF as well.

[Meta-Review · NeurIPS 2019]

This paper provides an outlier measure, along with an algorithm to estimate it, which handles heterogeneous data sets with attributes of different nature. This measure is based on information-theoretic intuitions of how difficult it is to identify or characterize the point, which can be expressed in terms of the sparsity of the containing sub-cube. Compared to previous work (such as Isolation Forest, and subsequent works), this is a richer notion that goes significantly beyond previous notions of "low density" regions for outlier detection, and also has benefits around interpretability.  The reviewers were unanimous in their vote to accept. Authors are encouraged to revise with respect to reviewer comments.